# Finding Second-Order Stationary Points in Nonconvex-Strongly-Concave Minimax Optimization

**Luo Luo**
School of Data Science
Fudan University
luoluo@fudan.edu.cn

**Yujun Li**
Noah's Ark Lab
Huawei Technologies Co., Ltd.
liyujun9@huawei.com

**Cheng Chen**[*]
School of Physical and Mathematical Sciences
Nanyang Technological University
cheng.chen@ntu.edu.sg

## Abstract

We study the smooth minimax optimization problem $\min_{\mathbf{x}} \max_{\mathbf{y}} f(\mathbf{x}, \mathbf{y})$, where $f$ is $\ell$-smooth, strongly-concave in $\mathbf{y}$ but possibly nonconvex in $\mathbf{x}$. Most of existing works focus on finding the first-order stationary points of the function $f(\mathbf{x}, \mathbf{y})$ or its primal function $P(\mathbf{x}) \triangleq \max_{\mathbf{y}} f(\mathbf{x}, \mathbf{y})$, but few of them focus on achieving second-order stationary points. In this paper, we propose a novel approach for minimax optimization, called Minimax Cubic Newton (MCN), which could find an $\left(\varepsilon, \kappa^{1.5}\sqrt{\rho\varepsilon}\right)$-second-order stationary point of $P(\mathbf{x})$ with calling $\mathcal{O}\left(\kappa^{1.5}\sqrt{\rho}\varepsilon^{-1.5}\right)$ times of second-order oracles and $\tilde{\mathcal{O}}\left(\kappa^{2}\sqrt{\rho}\varepsilon^{-1.5}\right)$ times of first-order oracles, where $\kappa$ is the condition number and $\rho$ is the Lipschitz continuous constant for the Hessian of $f(\mathbf{x}, \mathbf{y})$. In addition, we propose an inexact variant of MCN for high-dimensional problems to avoid calling expensive second-order oracles. Instead, our method solves the cubic sub-problem inexactly via gradient descent and matrix Chebyshev expansion. This strategy still obtains the desired approximate second-order stationary point with high probability but only requires $\tilde{\mathcal{O}}\left(\kappa^{1.5}\ell\varepsilon^{-2}\right)$ Hessian-vector oracle calls and $\tilde{\mathcal{O}}\left(\kappa^{2}\sqrt{\rho}\varepsilon^{-1.5}\right)$ first-order oracle calls. To the best of our knowledge, this is the first work that considers the non-asymptotic convergence behavior of finding second-order stationary points for minimax problems without the convex-concave assumptions.

## 1 Introduction

We consider minimax optimization of the form

$$\min_{\mathbf{x}\in\mathbb{R}^{d_x}} \max_{\mathbf{y}\in\mathbb{R}^{d_y}} f(\mathbf{x}, \mathbf{y}), \tag{1}$$

where $f(\mathbf{x}, \mathbf{y})$ is $\ell$-smooth, $\mu$-strongly-concave in $\mathbf{y}$, but possibly nonconvex in $\mathbf{x}$. Problem (1) can also be written as

$$\min_{\mathbf{x}\in\mathbb{R}^{d_x}} \left\{ P(\mathbf{x}) \triangleq \max_{\mathbf{y}\in\mathbb{R}^{d_y}} f(\mathbf{x}, \mathbf{y}) \right\}. \tag{2}$$

This framework covers a wide range of applications in machine learning such as regularized GAN [32], reinforcement learning [31], domain adaptation [9] and adversarial training [35].

---

[*]The corresponding author

36th Conference on Neural Information Processing Systems (NeurIPS 2022).

Most recent works focus on finding an $\varepsilon$-first-order stationary point (FSP) of $P(\mathbf{x})$. Lin et al. [20] showed that the vanilla gradient descent ascent (GDA) method could obtain an $\varepsilon$-FSP with $\mathcal{O}((\kappa^2\ell + \kappa\ell^2)\varepsilon^{-2})$ first-order oracle calls. This complexity can be reduced to $\tilde{\mathcal{O}}\left(\sqrt{\kappa}\ell\varepsilon^{-2}\right)$ by proximal iteration algorithms [21], which matches the gradient oracle lower bound for finding $\varepsilon$-FSP of $P(\mathbf{x})$ [13, 44]. The theory of first-order optimization for problem (1) has also been studied in stochastic settings [12, 15, 20, 24, 41, 42] and the block-wise setting [23]. However, the approximate FSP obtained by these algorithms cannot guarantee the local optimality since the primal function $P(\mathbf{x})$ could be nonconvex.

In this paper, we focus on finding a second-order stationary point (SSP) of $P(\mathbf{x})$ to capture the local optimal properties [8, 26]. Inspired by the success of second-order optimization in nonconvex minimization [1, 5, 6, 14, 18, 22, 28, 38, 45], we propose a novel method, called Minimax Cubic Newton (MCN), which runs cubic Newton update on $\mathbf{x}$ and maximizes the objective on $\mathbf{y}$ alternatively. This iteration scheme avoids getting stuck at an unexpected FSP. Specifically, we show MCN will converge to an $\left(\varepsilon, \kappa^{1.5}\sqrt{\rho\varepsilon}\right)$-SSP of $P(\mathbf{x})$ with $\mathcal{O}\left(\kappa^{1.5}\sqrt{\rho}\varepsilon^{-1.5}\right)$ number of iterations, where $\kappa$ is the condition number and $\rho$ is the Lipschitz continuous constant of $\nabla^2 f(\mathbf{x}, \mathbf{y})$. For high-dimensional problems, we also propose an efficient algorithm, called Inexact Minimax Cubic Newton (IMCN), which avoids the expensive second-order oracle calls. IMCN approximates the second-order information by matrix Chebyshev polynomial and solves the cubic regularized sub-problem inexactly. It only requires $\tilde{\mathcal{O}}\left(\kappa^{1.5}\ell\varepsilon^{-2}\right)$ Hessian-vector oracle calls and $\tilde{\mathcal{O}}\left(\kappa^2\sqrt{\rho}\varepsilon^{-1.5}\right)$ first-order oracle calls to find an $\left(\varepsilon, \kappa^{1.5}\sqrt{\rho\varepsilon}\right)$-SSP. Under mild strict saddle condition [3, 10, 11, 36, 37], the approximate SSP of $P(\mathbf{x})$ implies an approximate local minimax point of $f(\mathbf{x}, \mathbf{y})$ defined by Jin et al. [17], which successfully characterizes the local optimality for problem (1). To the best of our knowledge, this is the first work that considers non-asymptotic convergence behavior of finding SSP for minimax problems without convex-concave assumptions. We also conduct experiments on both synthetic function and the real application to validate our theoretical analysis. The empirical results show that the proposed algorithms significantly outperform the GDA method.

In a concurrent work, Chen et al. [7] also studied Problem (1) and proposed Cubic-GDA which is similar to our MCN algorithm. MCN has advantage on complexity of first-order oracles by a factor of $\sqrt{\kappa}$ since Cubic-GDA adopts GD to update $\mathbf{y}$ while MCN uses AGD instead. Chen et al. [7] mentioned that the cubic sub-problem can be efficiently solved by gradient-based algorithms, but they did not provide theoretical analysis for this inexact variant, which is more practical in high dimensional case. As a comparison, we provide the complexity of both Hessian-vector oracles and first-order oracles of our inexact algorithm IMCN.

## 2 Preliminaries

This section first presents the notations and assumptions for our settings. Then we introduce the background of local optimality for minimax optimization and some basic algorithms.

### 2.1 Notations and Assumptions

For a twice differentiable function $f(\mathbf{x}, \mathbf{y})$, its partial gradients with respect to $\mathbf{x}$ and $\mathbf{y}$ are denoted as $\nabla_x f(\mathbf{x}, \mathbf{y})$ and $\nabla_y f(\mathbf{x}, \mathbf{y})$ respectively. Its Hessian matrix at point $(\mathbf{x}, \mathbf{y})$ can be partitioned as $\nabla^2 f(\mathbf{x}, \mathbf{y}) = [\nabla^2_{xx} f(\mathbf{x}, \mathbf{y}), f(\mathbf{x}, \mathbf{y}); \nabla^2_{yx} f(\mathbf{x}, \mathbf{y}), \nabla^2_{yy} f(\mathbf{x}, \mathbf{y})]$, where $\nabla^2_{xx} f(\mathbf{x}, \mathbf{y}) \in \mathbb{R}^{d_x \times d_x}$, $\nabla^2_{xy} f(\mathbf{x}, \mathbf{y}) \in \mathbb{R}^{d_x \times d_y}$, $\nabla^2_{yx} f(\mathbf{x}, \mathbf{y}) \in \mathbb{R}^{d_y \times d_x}$ and $\nabla^2_{yy} f(\mathbf{x}, \mathbf{y}) \in \mathbb{R}^{d_y \times d_y}$. We also denote $\mathbf{H}(\mathbf{x}, \mathbf{y}) = \nabla^2_{xx} f(\mathbf{x}, \mathbf{y}) - \nabla^2_{xy} f(\mathbf{x}, \mathbf{y})(\nabla^2_{yy} f(\mathbf{x}, \mathbf{y}))^{-1}\nabla^2_{yx} f(\mathbf{x}, \mathbf{y})$ if $\nabla^2_{yy} f(\mathbf{x}, \mathbf{y})$ is invertible.

Given a symmetric matrix $\mathbf{A}$, we denote $\lambda_{\min}(\mathbf{A})$ as the smallest eigenvalue of $\mathbf{A}$. We use $\|\cdot\|_2$ to denote the spectral norm of matrices and Euclidean norm of vectors. We also denote the closed Euclidean ball with radius $r$ and center $\mathbf{x}^*$ as $\mathcal{B}(\mathbf{x}^*, r) = \{\mathbf{x} : \|\mathbf{x} - \mathbf{x}^*\|_2 \leq r\}$. Additionally, we use notation $\tilde{\mathcal{O}}(\cdot)$ to hide logarithmic terms in the complexity.

We suppose the objective function $f(\mathbf{x}, \mathbf{y})$ of Problem (1) satisfies the following assumptions.

**Assumption 1.** *The function $f(\mathbf{x}, \mathbf{y})$ has $\ell$-Lipschitz gradients, i.e., there exists a constant $\ell > 0$ such that $\|\nabla f(\mathbf{x}, \mathbf{y}) - \nabla f(\mathbf{x}', \mathbf{y}')\|_2^2 \leq \ell^2\left(\|\mathbf{x} - \mathbf{x}'\|_2^2 + \|\mathbf{y} - \mathbf{y}'\|_2^2\right)$ for any $\mathbf{x}, \mathbf{x}' \in \mathbb{R}^{d_x}$ and $\mathbf{y}, \mathbf{y}' \in \mathbb{R}^{d_y}$.*

**Assumption 2.** *The function $f(\mathbf{x}, \mathbf{y})$ has $\rho$-Lipschitz Hessian, i.e., there exists a constant $\rho > 0$ such that $\left\| \nabla^2 f(\mathbf{x}, \mathbf{y}) - \nabla^2 f(\mathbf{x}', \mathbf{y}') \right\|_2^2 \leq \rho^2 \left( \|\mathbf{x} - \mathbf{x}'\|_2^2 + \|\mathbf{y} - \mathbf{y}'\|_2^2 \right)$ for any $\mathbf{x}, \mathbf{x}' \in \mathbb{R}^{d_x}$ and $\mathbf{y}, \mathbf{y}' \in \mathbb{R}^{d_y}$.*

**Assumption 3.** *The function $f(\mathbf{x}, \mathbf{y})$ is $\mu$-strongly-concave in $\mathbf{y}$, i.e., there exists a constant $\mu > 0$ such that $f(\mathbf{x}, \mathbf{y}) \leq f(\mathbf{x}, \mathbf{y}') + \nabla_y f(\mathbf{x}, \mathbf{y})^\top (\mathbf{y} - \mathbf{y}') - \frac{\mu}{2} \|\mathbf{y} - \mathbf{y}'\|_2^2$ for any $\mathbf{x} \in \mathbb{R}^{d_x}$ and $\mathbf{y}, \mathbf{y}' \in \mathbb{R}^{d_y}$.*

**Assumption 4.** *The function $P(\mathbf{x}) \triangleq \max_{\mathbf{y} \in \mathbb{R}^{d_y}} f(\mathbf{x}, \mathbf{y})$ satisfies $P^* \triangleq \inf_{\mathbf{x} \in \mathbb{R}^{d_x}} P(\mathbf{x}) > -\infty$.*

**Definition 1.** *Under Assumption 1 and 3, we define the condition number of $f(\mathbf{x}, \mathbf{y})$ as $\kappa \triangleq \ell/\mu$.*

The assumptions of Lipschitz continuous gradient and strongly-concavity on $f$ indicate that the primal function $P(\mathbf{x}) \triangleq \max_{\mathbf{y} \in \mathbb{R}^{d_y}} f(\mathbf{x}, \mathbf{y})$ is well-defined and has Lipschitz continuous gradients as shown in Lemma 1.

**Lemma 1** ([20, Lemma 4.3]). *Suppose the objective function $f$ satisfies Assumptions 1 and 3, then the primal function $P(\mathbf{x}) \triangleq \max_{\mathbf{y} \in \mathbb{R}^{d_y}} f(\mathbf{x}, \mathbf{y})$ has $(\kappa + 1)\ell$-Lipschitz continuous gradients. Additionally, the function $\mathbf{y}^*(\mathbf{x}) = \arg\max_{\mathbf{y} \in \mathbb{R}^{d_y}} f(\mathbf{x}, \mathbf{y})$ is well-defined and $\kappa$-Lipschitz. We also have $\nabla P(\mathbf{x}) = \nabla_{\mathbf{x}} f(\mathbf{x}, \mathbf{y}^*(\mathbf{x}))$.*

Now we give the definitions of $\varepsilon$-FSP and $(\varepsilon, \delta)$-SSP as follows.

**Definition 2.** *Suppose the function $f(\mathbf{x}, \mathbf{y})$ satisfies Assumption 1 and 3, then we call $\mathbf{x}$ an $\varepsilon$-FSP of $P(\mathbf{x})$ if $\|\nabla P(\mathbf{x})\|_2 \leq \varepsilon$.*

**Definition 3.** *Suppose the function $f(\mathbf{x}, \mathbf{y})$ satisfies Assumption 1, 2 and 3, then we call $\mathbf{x}$ is an $(\varepsilon, \delta)$-SSP of $P(\mathbf{x})$ if $\|\nabla P(\mathbf{x})\|_2 \leq \varepsilon$ and $\nabla^2 P(\mathbf{x}) \succeq -\delta \mathbf{I}$.*

The following two lemmas provide the closed form of $\nabla^2 P(\mathbf{x})$ and its Lipschitz continuity.

**Lemma 2** ([34]). *Suppose the function $f(\mathbf{x}, \mathbf{y})$ satisfies Assumption 1, 2 and 3. We use the definition of $\mathbf{y}^*(\mathbf{x})$ in Lemma 1, then it holds that $\nabla^2 P(\mathbf{x}) = \mathbf{H}(\mathbf{x}, \mathbf{y}^*(\mathbf{x}))$.*

**Lemma 3.** *Under assumptions of Lemma 2, we have $\left\| \nabla^2 P(\mathbf{x}) - \nabla^2 P(\mathbf{x}') \right\|_2 \leq 4\sqrt{2}\kappa^3 \rho \|\mathbf{x} - \mathbf{x}'\|_2$ for any $\mathbf{x}$ and $\mathbf{x}'$ in $\mathbb{R}^{d_x}$.*

## 2.2 Local Optimality of Minimax Optimization

The Nash equilibrium is widely used in the study of convex-concave minimax optimization [40, 43], but it is intractable in general when the objective function $f(\mathbf{x}, \mathbf{y})$ is nonconvex in $\mathbf{x}$ or nonconcave in $\mathbf{y}$. For the general minimax problem, we introduce the local minimax point [17], which characterizes the optimality in two-player sequential games where players can only change their strategies locally.

**Definition 4** ([17]). *Given a differentiable function $f(\mathbf{x}, \mathbf{y}) : \mathbb{R}^{d_x} \times \mathbb{R}^{d_y} \to \mathbb{R}$ that is strongly-concave in $\mathbf{y}$, a point $(\mathbf{x}^*, \mathbf{y}^*) \in \mathbb{R}^{d_x} \times \mathbb{R}^{d_y}$ is called a local minimax point of $f$, if there exists $\delta_0 > 0$ and a function $h$ satisfying $h(\delta) \to 0$ as $\delta \to 0$, such that for any $\delta \in (0, \delta_0]$, $\mathbf{x} \in \mathcal{B}(\mathbf{x}^*, h(\delta))$ and $\mathbf{y} \in \mathbb{R}^{d_y}$, we have*

$$f(\mathbf{x}^*, \mathbf{y}) \leq f(\mathbf{x}^*, \mathbf{y}^*) \leq \max_{\mathbf{y}' \in \mathbb{R}^{d_y}} f(\mathbf{x}, \mathbf{y}'). \tag{3}$$

**Remark 1.** *The definition of local minimax point for general nonconvex-nonconcave [17] only requires (3) holds for any $\mathbf{y}$ and $\mathbf{y}'$ in a neighbour of $\mathbf{y}^*$, while the constraint on $\mathbf{y}$ and $\mathbf{y}'$ is unnecessary in our setting since we assume that $f(\mathbf{x}, \mathbf{y})$ is strongly-concave in $\mathbf{y}$.*

The local minimax point enjoys the following property.

**Lemma 4** ([17, Proposition 19]). *Suppose $f(\mathbf{x}, \mathbf{y})$ is twice differentiable, then any point $(\mathbf{x}^*, \mathbf{y}^*)$ satisfying $\nabla f(\mathbf{x}^*, \mathbf{y}^*) = \mathbf{0}$, $\nabla_{yy}^2 f(\mathbf{x}^*, \mathbf{y}^*) \prec \mathbf{0}$ and $\mathbf{H}(\mathbf{x}^*, \mathbf{y}^*) \succ \mathbf{0}$ is a local minimax point of $f(\mathbf{x}, \mathbf{y})$.*

Based on Lemma 4, we introduce the strict-minimax condition on $f$ which is an extension of strict-saddle condition for nonconvex minimization [3, 10, 11, 16, 36, 37].

**Definition 5.** *Under Assumption 1, 2 and 3, we say $f(\mathbf{x}, \mathbf{y})$ is $(\alpha, \beta, \gamma)$-strict-minimax for some $\alpha > 0$, $\beta > 0$ and $\gamma > 0$ if every $(\hat{\mathbf{x}}, \hat{\mathbf{y}}) \in \mathbb{R}^{d_x} \times \mathbb{R}^{d_y}$ satisfies at least one of the following three conditions: (a) $\|\nabla f(\hat{\mathbf{x}}, \hat{\mathbf{y}})\|_2 > \alpha$; (b) $\lambda_{\min}(\mathbf{H}(\hat{\mathbf{x}}, \hat{\mathbf{y}})) < -\beta$; (c) There exists a local minimax point $(\mathbf{x}^*, \mathbf{y}^*)$ such that $\|\hat{\mathbf{x}} - \mathbf{x}^*\|_2^2 + \|\hat{\mathbf{y}} - \mathbf{y}^*\|_2^2 \le \gamma^2$.*

Note that if $f$ is $(\alpha, \beta, \gamma)$-strict-minimax and there exists a point $\hat{\mathbf{x}}$ which is an $(\varepsilon, \delta)$-SSP of $P(\mathbf{x})$ with sufficient small $\varepsilon$ and $\delta$, then we can find $\hat{\mathbf{y}} \approx \arg\max_{\mathbf{y}} f(\hat{\mathbf{x}}, \mathbf{y})$ via running a first-order algorithm to minimize $-f(\hat{\mathbf{x}}, \cdot)$ and obtain $(\hat{\mathbf{x}}, \hat{\mathbf{y}})$ which is in a neighborhood of $(\mathbf{x}^*, \mathbf{y}^*)$. In other words, under the strict-minimax condition, we can reduce the task of finding approximate local minimax point of $f(\mathbf{x}, \mathbf{y})$ to finding an approximate SSP of $P(\mathbf{x})$. We will provide the formal statement in Section 3.2.

### 2.3 Accelerated Gradient Descent

Nesterov's accelerated gradient descent (AGD) is the optimal first-order algorithm for convex optimization [27, 29], which is widely used in minimax optimization algorithms [21, 39]. We describe the details of AGD for smooth and strongly-convex functions in Algorithm 1, which has the following convergence rate.

---

**Algorithm 1** $\mathrm{AGD}(h, \mathbf{y}_0, K, \eta, \theta)$

1: $\tilde{\mathbf{y}}_0 = \mathbf{y}_0$
2: **for** $k = 0, \ldots, K - 1$ **do**
3:     $\mathbf{y}_{k+1} = \tilde{\mathbf{y}}_k - \eta \nabla h(\tilde{\mathbf{y}}_k)$
4:     $\tilde{\mathbf{y}}_{k+1} = \mathbf{y}_{t+1} + \theta(\mathbf{y}_{k+1} - \mathbf{y}_k)$
5: **end for**
6: **Output:** $\mathbf{y}_K$

---

**Lemma 5** ([39, Lemma 2]). *Running Algorithm 1 on a $\ell_h$-smooth and $\mu_h$-strongly-convex objective function $h(\cdot)$ with parameters $\eta = 1/\ell_h$ and $\theta = \frac{\sqrt{\kappa_h} - 1}{\sqrt{\kappa_h} + 1}$ produces the output $\mathbf{y}_K$ satisfying $\|\mathbf{y}_K - \mathbf{y}^*\|_2^2 \le (\kappa_h + 1)(1 - \frac{1}{\sqrt{\kappa_h}})^K \|\mathbf{y}_0 - \mathbf{y}^*\|_2^2$, where $\mathbf{y}^* = \arg\min_y h(\mathbf{y})$ and $\kappa_h = \ell_h / \mu_h$.*

### 2.4 Cubic Regularized Newton

Cubic regularized Newton (CRN) is a classic algorithm for nonconvex minimization [5, 6, 27, 28, 38]. It solves the nonconvex minimization problem $\min_{\mathbf{x}} g(\mathbf{x})$ via the following update rules

$$\mathbf{s}_t = \arg\min_{\mathbf{s} \in \mathbb{R}^d} \nabla g(\mathbf{x}_t)^\top \mathbf{s} + \frac{1}{2} \mathbf{s}^\top \nabla^2 g(\mathbf{x}_t) \mathbf{s} + \frac{\rho_g}{6} \|\mathbf{s}\|_2^3, \qquad \mathbf{x}_{t+1} = \mathbf{x}_t + \mathbf{s}_t.$$

The CRN method could find an $(\varepsilon, \sqrt{\rho_g \varepsilon})$-SSP of $g(\mathbf{x})$ with $\mathcal{O}(\sqrt{\rho_g} \varepsilon^{-1.5})$ number of iterations, where $\rho_g$ is the Lipschitz continuous constant of the Hessian of $g(\mathbf{x})$.

## 3 Minimax Cubic Newton Algorithm

In this section, we propose our minimax cubic Newton algorithm and give its convergence results.

### 3.1 Minimax Cubic Newton Method

We present the details of Minimax Cubic Newton (MCN) method in Algorithm 2. In each round, the MCN algorithm performs following steps:

- Run AGD as presented in Algorithm 1 to estimate $\mathbf{y}_t \approx \mathbf{y}^*(\mathbf{x}_t) = \arg\max_{\mathbf{y}} f(\mathbf{x}_t, \mathbf{y})$.

- Compute the inexact first-order and second-order information of $P$ at $\mathbf{x}_t$ as

$$\nabla P(\mathbf{x}_t) \approx \mathbf{g}_t = \nabla_x f(\mathbf{x}_t, \mathbf{y}_t) \quad \text{and} \quad \nabla^2 P(\mathbf{x}_t) \approx \mathbf{H}_t = \mathbf{H}(\mathbf{x}_t, \mathbf{y}_t).$$

- Solve the following cubic regularized problem

$$\mathbf{s}_t^* = \arg\min_{\mathbf{s} \in \mathbb{R}^{d_x}} \left( \mathbf{g}_t^\top \mathbf{s} + \frac{1}{2} \mathbf{s}^\top \mathbf{H}_t \mathbf{s} + \frac{M}{6} \|\mathbf{s}\|_2^3 \right). \tag{4}$$

The expressions of $\mathbf{g}_t$ and $\mathbf{H}_t$ in the algorithm are inspired from Lemma 1 and 3. The smoothness of $\nabla P(\mathbf{x})$ and $\nabla^2 P(\mathbf{x})$ allow the total complexity of AGD iteration of the algorithm has the desired upper bound. Our theoretical analysis show the termination condition in Line 7 of Algorithm 2 can be attained in no more than $\mathcal{O}(\kappa^{1.5}\sqrt{\rho}\varepsilon^{-1.5})$ number of iterations. We also show that a small $\|\mathbf{s}_t^*\|_2$ will lead to a desired approximate second-order stationary point of $P(\mathbf{x})$. Hence, using AGD to optimize $f(\mathbf{x} + \mathbf{s}_t^*, \mathbf{y})$ with respect to $\mathbf{y}$ generates an approximate local minimax point of $f(\mathbf{x}, \mathbf{y})$.

---

**Algorithm 2** Minimax Cubic-Newton (MCN)

1: **Input:** $\mathbf{x}_0 \in \mathbb{R}^{d_x}$, $\mathbf{y}_{-1} = \mathbf{0}$, $T$, $\{K_t\}_{t=0}^T$, $\varepsilon$.
2: **for** $t = 0, \cdots T - 1$ **do**
3:    $\mathbf{y}_t = \mathrm{AGD}\left(-f(\mathbf{x}_t, \cdot), \mathbf{y}_{t-1}, K_t, \frac{1}{\ell}, \frac{\sqrt{\kappa}-1}{\sqrt{\kappa}+1}\right)$
4:    $\mathbf{g}_t = \nabla_x f(\mathbf{x}_t, \mathbf{y}_t)$, $\mathbf{H}_t = \mathbf{H}(\mathbf{x}_t, \mathbf{y}_t)$
5:    $\mathbf{s}_t^* = \underset{\mathbf{s} \in \mathbb{R}^{d_x}}{\arg\min}\left(\mathbf{g}_t^\top \mathbf{s} + \frac{1}{2}\mathbf{s}^\top \mathbf{H}_t \mathbf{s} + \frac{M}{6}\|\mathbf{s}\|_2^3\right)$
6:    **if** $\|\mathbf{s}_t^*\|_2 \le \frac{1}{2}\sqrt{\varepsilon/M}$ **then** break
7:    $\mathbf{x}_{t+1} = \mathbf{x}_t + \mathbf{s}_t^*$
8: **end for**
9: **Output:** $\hat{\mathbf{x}} = \mathbf{x}_t + \mathbf{s}_t^*$

---

### 3.2 Complexity Analysis for MCN

In this section, we let $M = 4\sqrt{2}\kappa^3\rho$ for MCN. Our analysis for MCN algorithm contains three parts:

1. We follow Tripuraneni et al. [38]'s idea to show that our algorithm with sufficient accurate gradient and Hessian estimator of $P(\mathbf{x})$ requires no more than $T = \mathcal{O}\left(\kappa^{1.5}\sqrt{\rho}\varepsilon^{-1.5}\right)$ rounds of iterations to achieve an $\left(\varepsilon, \kappa^{1.5}\sqrt{\rho\varepsilon}\right)$-SSP of $P(\mathbf{x})$.

2. We prove the AGD step in line 3 requires at most $\tilde{\mathcal{O}}\left(\kappa^2\sqrt{\rho}\varepsilon^{-1.5}\right)$ gradient calls in total.

3. The last part shows we can achieve an approximate local minimax point of $f(\mathbf{x}, \mathbf{y})$ from an $\left(\varepsilon, \kappa^{1.5}\sqrt{\rho\varepsilon}\right)$-second-order stationary point of $P(\mathbf{x})$ under strict-saddle condition.

**Cubic Newton Iteration on $P(\mathbf{x})$** The procedure of our Algorithm 2 can be regarded as applying cubic Newton method to minimize nonconvex function $P(\mathbf{x})$, but using inexact first-order and second-order information. We consider the following conditions on the inexact gradient and Hessian, which will hold if we run AGD with enough number of iterations.

**Assumption 5.** *Suppose the estimators* $\mathbf{g}_t \in \mathbb{R}^{d_x}$ *and* $\mathbf{H}_t \in \mathbb{R}^{d_x \times d_x}$ *satisfy conditions* $\|\nabla P(\mathbf{x}_t) - \mathbf{g}_t\|_2 \le C_g\varepsilon$ *and* $\|\nabla^2 P(\mathbf{x}_t) - \mathbf{H}_t\|_2 \le C_H\sqrt{M\varepsilon}$ *for some* $C_g > 0$ *and* $C_H > 0$.

The following lemma implies the analysis of MCN algorithm only needs to focus on the case of each $\|\mathbf{s}_t^*\|_2$ is large, otherwise we have already find $\mathbf{x}_{t+1}$ which a desired approximate SSP of $P(\mathbf{x})$.

**Lemma 6.** *Under Assumption 5 with* $C_g = 1/192$ *and* $C_H = 1/48$, *if* $\mathbf{x}_{t+1}$ *from Algorithm 2 is not an* $\left(\varepsilon, \sqrt{M\varepsilon}\right)$-SSP *of* $P(\mathbf{x})$, *we have* $\|\mathbf{s}_t^*\|_2 \ge \frac{1}{2}\sqrt{\varepsilon/M}$.

By Lemma 6, we can show that MCN with sufficient accurate gradient and Hessian estimators can find an $\left(\varepsilon, \sqrt{M\varepsilon}\right)$-SSP of $P(\mathbf{x})$ with $T = \mathcal{O}\left(\kappa^{1.5}\sqrt{\rho}\varepsilon^{-1.5}\right)$ number of iterations as follows.

**Theorem 1.** *Under Assumption 1-4, we run Algorithm 2 with* $T = \left\lceil 192(P(\mathbf{x}_0) - P^*)\sqrt{M}\varepsilon^{-1.5}\right\rceil + 1$; *and further suppose* $K_t$ *is sufficient large so that results* $\mathbf{g}_t$ *and* $\mathbf{H}_t$ *satisfy Assumption 5 with* $C_g = 1/192$ *and* $C_H = 1/48$. *Then the output* $\hat{\mathbf{x}}$ *is an* $\left(\varepsilon, \sqrt{M\varepsilon}\right)$-SSP *of* $P(\mathbf{x})$.

**Total Complexity of AGD** Note that MCN (Algorithm 2) applies AGD to maximize $f(\mathbf{x}_t, \cdot)$ by using $\mathbf{y}_{t-1}$ as initialization at the $t$-th round. With such initialization, the following theorem provides the upper bound of total number of gradient calls required by AGD and guarantees that $\mathbf{g}_t$ and $\mathbf{H}_t$ in the MCN algorithm satisfy Assumption 5.

**Theorem 2.** *Under Assumption 1-4, we run Algorithm 2 with* $K_0 = \left\lceil 2\sqrt{\kappa}\log\left(\frac{\sqrt{\kappa}+1}{\tilde{\varepsilon}}\|\mathbf{y}^*(\mathbf{x}_0)\|_2\right)\right\rceil$ *and* $K_t = \left\lceil 2\sqrt{\kappa}\log\left(\frac{\sqrt{\kappa}+1}{\tilde{\varepsilon}}(\tilde{\varepsilon} + \kappa\|\mathbf{s}_{t-1}^*\|_2)\right)\right\rceil$ *for* $t \ge 1$, *where* $\tilde{\varepsilon} = \min\left\{C_g\varepsilon/\ell, C_H\sqrt{M\varepsilon}/\rho\right\}$ *and* $\mathbf{s}_0 = \mathbf{0}$. *Then all* $\mathbf{g}_t$ *and* $\mathbf{H}_t$ *satisfy the condition of Assumption 5. We also have*

$$\sum_{t=0}^T K_t \le T + 1 + \frac{2\sqrt{\kappa}T}{3}\left[3\log\left(\frac{\sqrt{\kappa}+1}{\tilde{\varepsilon}}\|\mathbf{y}^*(\mathbf{x}_0)\|_2\right) + \log\left(8(\kappa+1)^{1.5} + \frac{8\kappa^3(\kappa+1)^{1.5}}{T\tilde{\varepsilon}^3}\sum_{t=1}^T \|\mathbf{s}_{t-1}^*\|_2^3\right)\right].$$

| **Algorithm 3** Inexact Minimax Cubic-Newton | **Algorithm 4** Cubic-Solver |
|---|---|
| 1: **Input:** $\mathbf{x}_0 \in \mathbb{R}^{d_x}$, $\mathbf{y}_{-1} = \mathbf{0}$, $T$, $\{K_t\}_{t=0}^T$, $\varepsilon$ | 1: **Input:** $\mathbf{g}$, $\mathbf{H}$, $\sigma$, $\mathcal{K}(\varepsilon, \delta')$ |
| 2: $c_k = \frac{2}{\sqrt{\ell\mu}} \left( \frac{\sqrt{\mu/\ell}-1}{\sqrt{\mu/\ell}+1} \right)^k$ | 2: **if** $\|\mathbf{g}\|_2 \geq L^2/M$ **then** |
| 3: **for** $t = 0, \cdots$ **do** | 3: $R_C = -\frac{\mathbf{g}^\top \mathbf{H} \mathbf{g}}{M \|\mathbf{g}\|_2^2} + \sqrt{\left( \frac{\mathbf{g}^\top \mathbf{H} \mathbf{g}}{M \|\mathbf{g}\|_2^2} \right)^2 + \frac{2\|\mathbf{g}\|_2}{M}}$ |
| 4: $\quad \mathbf{y}_t = \mathrm{AGD}\big( -f(\mathbf{x}_t, \cdot), \mathbf{y}_{t-1}, K_t, \frac{1}{\ell}, \frac{\sqrt{\kappa}-1}{\sqrt{\kappa}+1} \big)$ | |
| 5: $\quad \mathbf{g}_t = \nabla_x f(\mathbf{x}_t, \mathbf{y}_t)$ | 4: $\quad \hat{\mathbf{s}} = -R_C \mathbf{g}/\|\mathbf{g}\|_2$ |
| 6: $\quad \mathbf{Z}_t = -\frac{2}{\ell-\mu} \big( \nabla_{yy}^2 f(\mathbf{x}_t, \mathbf{y}_t) + \frac{\ell+\mu}{2}\mathbf{I} \big)$ | 5: **else** |
| 7: $\quad$ Compute $\mathbf{H}_t$ as equation (5) | 6: $\quad \mathbf{s}_0 = \mathbf{0}$, $\eta = 1/(20L)$ |
| 8: $\quad (\mathbf{s}_t, \Delta_t) = \mathrm{Cubic\text{-}Solver}(\mathbf{g}_t, \mathbf{H}_t, \sigma, \mathcal{K}(\varepsilon, \delta'))$ | 7: $\quad \tilde{\mathbf{g}} = \mathbf{g} + \sigma\boldsymbol{\zeta}$ for $\boldsymbol{\zeta} \sim \mathrm{Uniform}(\mathbb{S}^{d-1})$ |
| 9: $\quad \mathbf{x}_{t+1} = \mathbf{x}_t + \mathbf{s}_t$ | 8: $\quad$ **for** $k = 0, \cdots, \mathcal{K}(\varepsilon, \delta')$ **do** |
| 10: $\quad$ **if** $\Delta_t > -\frac{1}{128}\sqrt{\varepsilon^3/M}$ **then** | 9: $\quad\quad \mathbf{s}_{k+1} = \mathbf{s}_k - \eta \big( \tilde{\mathbf{g}} + \mathbf{H}\mathbf{s}_k + \frac{M}{2}\|\mathbf{s}_k\|_2 \mathbf{s}_k \big)$ |
| 11: $\quad\quad \hat{\mathbf{s}} = \mathrm{Final\text{-}Cubic\text{-}Solver}(\mathbf{g}_t, \mathbf{H}_t, \varepsilon)$ | 10: $\quad$ **end for** |
| 12: $\quad\quad \mathbf{x}_{t+1} = \mathbf{x}_t + \hat{\mathbf{s}}$ | 11: $\quad \hat{\mathbf{s}} = \mathbf{s}_{\mathcal{K}(\varepsilon, \delta')}$ |
| 13: $\quad\quad$ **break** | 12: **end if** |
| 14: $\quad$ **end if** | 13: **Output:** $\hat{\mathbf{s}}$ and $\Delta = \mathbf{g}^\top \hat{\mathbf{s}} + \frac{1}{2}\hat{\mathbf{s}}^\top \mathbf{H}\hat{\mathbf{s}} + \frac{M}{6}\|\hat{\mathbf{s}}\|_2^3$ |
| 15: **end for** | |
| 16: **Output:** $\hat{\mathbf{x}} = \mathbf{x}_{t+1}$ | |

Combining Theorem 1 and Theorem 2, we can obtain the total number of gradient oracle calls, Hessian (inverse) oracle calls and exact cubic sub-problem solver calls as follows.

**Corollary 1.** *Under Assumption 1-4, running Algorithm 2 with* $T = \lceil 192(P(\mathbf{x}_0) - P^*)\sqrt{M}\varepsilon'^{-1.5} \rceil + 1$, $K_0 = \lceil 2\sqrt{\kappa} \log\big( \frac{\sqrt{\kappa}+1}{\tilde{\varepsilon}'} \|\mathbf{y}^*(\mathbf{x}_0)\|_2 \big) \rceil$ *and* $K_t = \lceil 2\sqrt{\kappa} \log\big( \frac{\sqrt{\kappa}+1}{\tilde{\varepsilon}'}(\tilde{\varepsilon}' + \kappa \|\mathbf{s}_{t-1}\|_2) \big) \rceil$ *for* $t \geq 1$, *where* $\tilde{\varepsilon}' = \min\big\{ C_g \varepsilon'/\ell, C_H \sqrt{M}\varepsilon'/\rho \big\}$, $\mathbf{s}_0 = \mathbf{0}$ *and* $\varepsilon' = 2^{-2.5}\varepsilon$, *then the output* $\hat{\mathbf{x}}$ *is an* $\big(\varepsilon, \kappa^{1.5}\sqrt{\rho\varepsilon}\big)$-SSP *of* $P(\mathbf{x})$ *and the number of gradient oracle calls is at most* $\tilde{\mathcal{O}}\big(\kappa^2 \sqrt{\rho}\varepsilon^{-1.5}\big)$. *The total number of Hessian (inverse) oracle calls and exact cubic sub-problem solver calls is at most* $\mathcal{O}\big(\sqrt{\rho}\kappa^{1.5}\varepsilon^{-1.5}\big)$.

Note that MCN method needs to construct the Hessian estimator $\mathbf{H}_t = \mathbf{H}(\mathbf{x}_t, \mathbf{y}_t)$ and solves the cubic regularized sub-problem (4) in each round. Constructing $\mathbf{H}_t$ requires calling the second-order oracle at $(\mathbf{x}_t, \mathbf{y}_t)$ and taking $\mathcal{O}\big(d_y^3 + d_x d_y^2 + d_x^2 d_y\big)$ flops for matrix multiplication and inversion. Solving sub-problem (4) requires $\mathcal{O}\big(d_x^3\big)$ flops for matrix factorization or inversion [5, 6]. Hence, besides the gradient calls from AGD step, MCN requires $\mathcal{O}\big(d_x^3 + d_y^3\big)$ time complexity in each round and its space complexity is $\mathcal{O}\big(d_x^2 + d_y^2\big)$.

**Approximate Local Minimax Point** Under the strict-minimax condition, we can find an approximate local minimax point by performing an additional AGD procedure on the output of MCN.

**Corollary 2.** *Suppose* $f(\mathbf{x}, \mathbf{y})$ *is* $(\alpha, \beta, \gamma)$-strict-minimax *and satisfies Assumption 1-4. Let* $\hat{\mathbf{x}}$ *be the output of running Algorithm 2 with the setting of Corollary 1 and* $\varepsilon = \min\big\{\alpha/3, \beta^2/(8\kappa^3\rho)\big\}$. *Let* $\hat{\mathbf{y}} = \mathrm{AGD}\big( -f(\hat{\mathbf{x}}, \cdot), \mathbf{y}_t, \hat{K}, \frac{1}{\ell}, \frac{\sqrt{\kappa}-1}{\sqrt{\kappa}+1} \big)$ *where* $t$ *corresponds to the last iteration of Algorithm 2 such that* $\hat{\mathbf{x}} = \mathbf{x}_t + \mathbf{s}_t^*$ *and* $\hat{K} = \sqrt{\kappa} \log\big( \min\big\{\frac{\alpha}{2\ell}, \frac{\beta}{8\kappa^2\rho}\big\} / \big(\sqrt{\kappa+1}\big(\tilde{\varepsilon} + \frac{\kappa}{2^{2.25}}\sqrt{\frac{\varepsilon}{M}}\big)\big) \big)$. *Then there exists a local minimax point* $(\mathbf{x}^*, \mathbf{y}^*)$ *of* $f(\mathbf{x}, \mathbf{y})$ *such that* $\|\mathbf{x}^* - \hat{\mathbf{x}}\|_2^2 + \|\mathbf{y}^* - \hat{\mathbf{y}}\|_2^2 \leq \gamma^2$.

## 4 Inexact Minimax Cubic Newton Algorithm

In this section, we proposed an efficient algorithm called inexact minimax cubic Newton (IMCN), which avoids any operation related to the second-order oracle and only requires $\tilde{\mathcal{O}}\big(\kappa^{1.5}\ell\varepsilon^{-2}\big)$ Hessian-vector product calls and $\tilde{\mathcal{O}}\big(\kappa^2 \sqrt{\rho}\varepsilon^{-1.5}\big)$ gradient calls in total to find $\big(\varepsilon, \kappa^{1.5}\sqrt{\rho\varepsilon}\big)$-SSP of $P(\mathbf{x})$. Since the Hessian-vector products can be computed as fast as gradients [30, 33], IMCN is much more efficient than MCN in high-dimensional cases.

We present the details of IMCN in Algorithm 3. Unlike MCN which solves problem (4) exactly, IMCN uses gradient-based cubic sub-problem solver (Algorithm 4) [4] to compute

$$\mathbf{s}_t \approx \arg\min_{\mathbf{s}\in\mathbb{R}^{d_x}} m_t(\mathbf{s}) \triangleq \mathbf{g}_t^\top \mathbf{s} + \frac{1}{2}\mathbf{s}^\top \mathbf{H}_t \mathbf{s} + \frac{M}{6}\|\mathbf{s}\|_2^3.$$

If the condition in line 10 of Algorithm 3 holds, the point $\mathbf{x}_t + \mathbf{s}_t^*$ should be a desired approximate second-order stationary point. Due to $\mathbf{s}_t^*$ is hard to obtain, we introduce additional gradient descent steps (Algorithm 5) to approximate it by $\hat{\mathbf{s}}$ and use $\mathbf{x}_t + \hat{\mathbf{s}}$ as the final output.

---

**Algorithm 5** Final-Cubic-Solver

1: **Input: g, H,** $\varepsilon$
2: $\mathbf{s}_0 = \mathbf{0}$, $\mathbf{g}_0 = \mathbf{g}$, $\eta = 1/(20L)$
3: **for** $t = 0, \dots$ **do**
4:    **if** $\|\mathbf{g}_t\|_2 \leq \varepsilon/2$ **then break**
5:    $\mathbf{s}_{t+1} = \mathbf{s}_t - \eta\mathbf{g}_t$
6:    $\mathbf{g}_{t+1} = \mathbf{g} + \mathbf{H}\mathbf{s}_{t+1} + \frac{M}{2}\|\mathbf{s}_{t+1}\|_2 \mathbf{s}_{t+1}$
7: **end for**
8: **Output:** $\mathbf{s}_t$

---

The design and the convergence analysis of IMCN is more challenging than existing inexact cubic Newton algorithms for minimization problems [18, 38] since the Hessian estimator in IMCN has a more complicated structure. To address this issue, we approximate the Hessian $\nabla^2 P(\mathbf{x}_t)$ by $\mathbf{H}_t$ as

$$\mathbf{H}_t = \nabla^2_{xx}f(\mathbf{x}_t, \mathbf{y}_t) + \nabla^2_{xy}f(\mathbf{x}_t, \mathbf{y}_t)\mathbf{C}_t\nabla^2_{yx}f(\mathbf{x}_t, \mathbf{y}_t), \tag{5}$$

where $\mathbf{C}_t = \frac{c_0}{4\ell}\mathbf{I} + \frac{1}{2\ell}\sum_{k=1}^{K'} c_k \mathbf{T}_k(\mathbf{Z}_t)$ and $\mathbf{Z}_t = \frac{4\ell}{\ell-\mu}\left(-\frac{1}{2\ell}\nabla^2_{yy}f(\mathbf{x}_t, \mathbf{y}_t) - \frac{\ell+\mu}{4\ell}\mathbf{I}\right)$. Here $\mathbf{T}_k(\cdot)$ is the matrix Chebyshev polynomials (shown in Section 4.1) leading to $\mathbf{C}_t \approx -\left(\nabla^2_{yy}(\mathbf{x}_t, \mathbf{y}_t)\right)^{-1}$. Note that we never construct matrix $\mathbf{H}_t$ explicitly in implementation because all operations related to $\mathbf{H}_t$ can be reduced to compute Hessian-vector products, which avoid any second-order oracle calls or matrix factorization/inversion.

Then we provide the convergence analysis for the IMCN algorithm. Throughout this section, we let $L = 2\kappa\ell$ and $M = 4\sqrt{2}\kappa^3\rho$ be the Lipschitz continuous constants of $\nabla P(x)$ and $\nabla^2 P(x)$. We suppose $\varepsilon \leq L^2/M$, otherwise, the second-order condition $\nabla^2 P(\mathbf{x}) \succeq -\sqrt{M\varepsilon}\,\mathbf{I}$ always holds and we only need to use gradient methods [20, 21] to find first-order stationary point.

## 4.1 Approximating Hessian by Matrix Chebyshev Polynomials

We first show the error bound of matrix inverse approximation via matrix Chebyshev polynomials.

**Lemma 7.** *Suppose symmetric matrix* $\mathbf{X} \in \mathbb{R}^{d\times d}$ *satisfies* $\mu'\mathbf{I} \preceq \mathbf{X} \preceq \ell'\mathbf{I}$ *with* $0 < \mu' \leq \ell' < 1$, *then we have* $\left\|\mathbf{X}^{-1} - \left(\frac{c_0}{2}\mathbf{I} + \sum_{k=1}^{K'} c_k \mathbf{T}_k(\mathbf{Z}')\right)\right\|_2 \leq \frac{\sqrt{\ell'/\mu'}-1}{\sqrt{\ell'\mu'}}\left(1 - 2/(\sqrt{\ell'/\mu'}+1)\right)^{K'}$ *where* $\mathbf{Z}' = \frac{2}{\ell'-\mu'}\left(\mathbf{X} - \frac{\ell'+\mu'}{2}\mathbf{I}\right)$, $c_k = \frac{2}{\sqrt{\ell'\mu'}}\left(\frac{\sqrt{\mu'/\ell'}-1}{\sqrt{\mu'/\ell'}+1}\right)^k$, *and* $\mathbf{T}_k(\cdot)$ *are matrix Chebyshev polynomials with* $T_0(\mathbf{Z}') = \mathbf{I}$, $T_1(\mathbf{Z}') = \mathbf{Z}'$ *and* $\mathbf{T}_k(\mathbf{Z}') = 2\mathbf{Z}'\mathbf{T}_{k-1}(\mathbf{Z}') - \mathbf{T}_{k-2}(\mathbf{Z}')$ *for* $k \geq 2$.

Based on Lemma 7, we can bound the approximation error of the Hessian estimator $\mathbf{H}_t$ as follows.

**Lemma 8.** *Using the notation of Algorithm 3, we have*

$$\left\|\nabla^2_{xx}P(\mathbf{x}_t) - \mathbf{H}_t\right\|_2 \leq 3\rho\kappa^2\sqrt{\kappa+1}\left(1 - \frac{1}{\sqrt{\kappa}}\right)^{K_t/2}\|\mathbf{y}_{t-1} - \mathbf{y}^*(\mathbf{x}_t)\|_2 + \kappa\ell\left(1 - \frac{2}{\sqrt{\kappa}+1}\right)^{K'}.$$

Lemma 8 means using AGD with $K_t = \mathcal{O}\left(\sqrt{\kappa}\log(\kappa\rho/\varepsilon_H)\right) = \tilde{\mathcal{O}}\left(\sqrt{\kappa}\right)$[2] and the number of terms for Chebyshev polynomials with $K' = \mathcal{O}\left(\sqrt{\kappa}\log(\kappa\ell/\varepsilon_H)\right) = \tilde{\mathcal{O}}\left(\sqrt{\kappa}\right)$ could achieve $\mathbf{H}_t$ with $\left\|\nabla^2 P(\mathbf{x}_t) - \mathbf{H}_t\right\|_2 \leq \varepsilon_H$ for any $\varepsilon_H > 0$.

In the implementation of IMCN, all operations related to $\mathbf{H}_t$ can be viewed as computing Hessian-vector products. Actually, we can obtain $\mathbf{H}_t\mathbf{u}'$ with $\mathcal{O}(K') = \tilde{\mathcal{O}}(\sqrt{\kappa})$ Hessian-vector calls for any $\mathbf{u}' \in \mathbb{R}^{d_x}$, which avoids $\mathcal{O}\left(d_x^2 + d_y^2\right)$ space to keep Hessian matrices. The detailed implementation is presented in Appendix E.

---

[2]Rigorously speaking, the term $\log(\|\mathbf{y}_{t-1} - \mathbf{y}^*(\mathbf{x}_t)\|_2)$ also should be considered into the total complexity, which will be discussed in later sections.

## 4.2 Complexity Analysis for IMCN

The IMCN method calls a gradient-based sub-problem solver (Algorithm 4) to optimize the following cubic regularized problem [4, 38] in each iteration

$$\min_{\mathbf{s}\in\mathbb{R}^{d_x}} \tilde{m}_t(\mathbf{s}) \triangleq \mathbf{g}_t^\top\mathbf{s} + \frac{1}{2}\mathbf{s}^\top\mathbf{H}_t\mathbf{s} + \frac{M}{6}\|\mathbf{s}\|_2^3. \tag{6}$$

It requires at most $\mathcal{K}(\varepsilon, \delta') = \mathcal{O}\big(\frac{L}{\sqrt{M\varepsilon}}\big(\log\big(\frac{\sqrt{d_x}}{\delta'}\big) + \log\big(\frac{L+\sqrt{\rho\varepsilon}}{\sqrt{M\varepsilon}}\big)\big)\big)$ number of iterations to achieve an approximate solution with enough accuracy. The detailed analysis for the complexity of Algorithm 4 is deferred to appendix D.

Then we bound the total number of iterations for Algorithm 3.

**Theorem 3.** *Running Algorithm 3 with $C_\sigma = 1/4$, $\delta' = \delta/T$, $T = \lceil 626(P(\mathbf{x}_0) - P^*)\sqrt{M}\varepsilon^{-1.5}\rceil$ and sufficient large $K_t$ and $K'$ such that Assumption 5 holds with $C_g = 1/240$ and $C_H = 1/200$. Then the condition $\Delta_t \geq -\frac{1}{128}\sqrt{\varepsilon^3/M}$ in line 11 must hold in no more than $T = \mathcal{O}\big(\kappa^{1.5}\sqrt{\rho}\varepsilon^{-1.5}\big)$ iterations; and the output $\hat{\mathbf{x}}$ is an $(\varepsilon, 2\kappa^{1.5}\sqrt{\rho\varepsilon})$-SSP with probability $1-\delta$.*

We also bound the number of gradient calls from AGD procedure in Algorithm 3 as follows.

**Theorem 4.** *Under the setting of Theorem 3, if we run Algorithm 3 with*

$$K' = \left\lceil \frac{\sqrt{\kappa}+1}{2}\log\left(\frac{\kappa\ell}{2\min\{C_H\sqrt{M\varepsilon}, \varepsilon_H L\}}\right)\right\rceil \text{ and } K_t = \begin{cases} \left\lceil 2\sqrt{\kappa}\log\left(\frac{\sqrt{\kappa}+1}{\tilde{\varepsilon}}\|\mathbf{y}^*(\mathbf{x}_0)\|_2\right)\right\rceil, & t = 0 \\ \left\lceil 2\sqrt{\kappa}\log\left(\frac{\sqrt{\kappa}+1}{\tilde{\varepsilon}}\big(\tilde{\varepsilon} + \kappa\|\mathbf{s}_{t-1}\|_2\big)\right)\right\rceil, & t \geq 1 \end{cases}$$

*where $\tilde{\varepsilon} = \min\{C_g\varepsilon/\ell, \min\{C_H\sqrt{M\varepsilon}, \varepsilon_H L\}/(6\rho\kappa^2)\}$ and $\mathbf{s}_0 = \mathbf{0}$, then it holds that $\|\nabla P(\mathbf{x}_t) - \mathbf{g}_t\|_2 \leq C_g\varepsilon$, $\|\nabla^2 P(\mathbf{x}_t) - \mathbf{H}_t\|_2 \leq \min\{C_H\sqrt{M\varepsilon}, \varepsilon_H L\}$ and*

$$\sum_{t=0}^T K_t \leq T+1+\frac{2\sqrt{\kappa}T}{3}\left[\frac{3}{T}\log\left(\frac{\sqrt{\kappa+1}}{\tilde{\varepsilon}}\|\mathbf{y}^*(\mathbf{x}_0)\|_2\right)+\log\left(8(\kappa+1)^{1.5} + \frac{8\kappa^3(\kappa+1)^{1.5}}{T\tilde{\varepsilon}^3}\sum_{t=1}^T\|\mathbf{s}_{t-1}\|_2^3\right)\right].$$

Note that the value of $K' = \tilde{\mathcal{O}}(\sqrt{\kappa})$ corresponds to the number of Hessian vector calls for each iteration of cubic sub-problem solver (Algorithm 4 and 5). Combining Theorem 3, Theorem 4 and the value of $\mathcal{K}(\varepsilon, \delta')$, we obtain the main result for Algorithm 3 as follows.

**Corollary 3.** *Under Assumption 1-4, if we run Algorithm 3 with $C_\sigma = 1/4$, $\delta' = \delta/T$,*

$$K' = \left\lceil \frac{\sqrt{\kappa}+1}{2}\log\left(\frac{\kappa\ell}{2\min\{C_H\sqrt{M\varepsilon}, \frac{3L}{25}\}}\right)\right\rceil \text{ and } K_t = \begin{cases} \left\lceil 2\sqrt{\kappa}\log\left(\frac{\sqrt{\kappa}+1}{\tilde{\varepsilon}}\|\mathbf{y}^*(\mathbf{x}_0)\|_2\right)\right\rceil, & t = 0 \\ \left\lceil 2\sqrt{\kappa}\log\left(\frac{\sqrt{\kappa}+1}{\tilde{\varepsilon}}\big(\tilde{\varepsilon} + \kappa\|\mathbf{s}_{t-1}\|_2\big)\right)\right\rceil, & t \geq 1 \end{cases}$$

*where $T = \lceil 626(P(\mathbf{x}_0) - P^*)\sqrt{M}\varepsilon'^{-1.5}\rceil$, $C_g = 1/240$, $C_H = 1/200$, $\mathbf{s}_0 = \mathbf{0}$, $\tilde{\varepsilon}' = \varepsilon/4$ and $\tilde{\varepsilon} = \min\{C_g\varepsilon/(4\ell), \min\{C_H\sqrt{M\varepsilon}, \frac{3L}{25}\}/(12\rho\kappa^2)\}$, then the output $\hat{\mathbf{x}}$ is an $(\varepsilon, \kappa^{1.5}\sqrt{\rho\varepsilon})$-second-order stationary point of $P(\mathbf{x})$ with probability $1-\delta$ and the number of gradient calls is at most $\tilde{\mathcal{O}}\big(\kappa^2\sqrt{\rho}\varepsilon^{-1.5}\big)$. The number of Hessian-vector product calls is at most $\mathcal{O}\big(\kappa^{1.5}\ell\varepsilon^{-2}\big)$.*

Following the analysis of Corollary 2, we can also achieve an approximate local minimax point by running AGD based on the $\hat{\mathbf{x}}$ obtained from Algorithm 3.

## 5 Experiments

In this section, we conduct empirical studies for our methods against the classical GDA algorithm [20] on both synthetic problem and real-world application.

### 5.1 Synthetic Minimax Problem

We construct the following nonconvex-strongly-concave minimax problem:

$$\min_{\mathbf{x}\in\mathbb{R}^3}\max_{\mathbf{y}\in\mathbb{R}^2} f(\mathbf{x}, \mathbf{y}) = w(x_3) - \frac{y_1^2}{40} + x_1 y_1 - \frac{5y_2^2}{2} + x_2 y_2, \tag{7}$$

where $\mathbf{x} = [x_1, x_2, x_3]^\top$, $\mathbf{y} = [y_1, y_2]^\top$ and $w(\cdot)$ is the W-shaped scalar function [38] whose exact form is shown in Appendix G.1. It is easy to verify that the problem has an strict saddle point at $(\mathbf{x}_0, \mathbf{y}_0) = ([0, 0, 0]^\top, [0, 0]^\top)$.

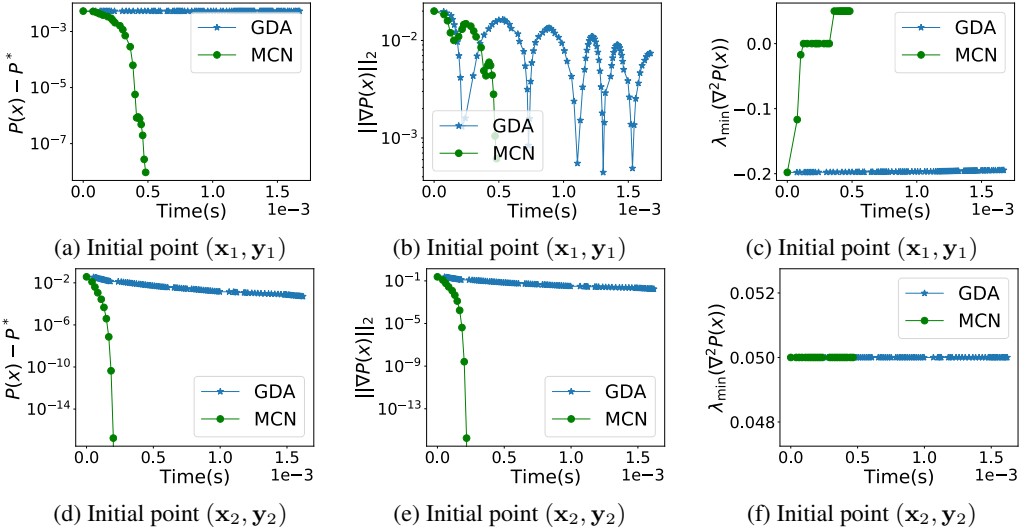

(a) Initial point $(\mathbf{x}_1, \mathbf{y}_1)$        (b) Initial point $(\mathbf{x}_1, \mathbf{y}_1)$        (c) Initial point $(\mathbf{x}_1, \mathbf{y}_1)$

(d) Initial point $(\mathbf{x}_2, \mathbf{y}_2)$        (e) Initial point $(\mathbf{x}_2, \mathbf{y}_2)$        (f) Initial point $(\mathbf{x}_2, \mathbf{y}_2)$

Figure 1: We present comparisons of error, gradient norms and Hessian minimum eigenvalues on the synthetic problem. Figure (a), (b) and (c) shows the results with initial point $(\mathbf{x}_1, \mathbf{y}_1)$. Figure (d), (e) and (f) shows the results with initial point $(\mathbf{x}_2, \mathbf{y}_2)$.

We conduct experiments on problem (7) with two different initial points

$$(\mathbf{x}_1, \mathbf{y}_1) = \left([10^{-3}, 10^{-3}, 10^{-3}]^\top, [0, 0]^\top\right) \quad \text{and} \quad (\mathbf{x}_2, \mathbf{y}_2) = \left([0, 0, 1]^\top, [0, 0]^\top\right).$$

Notice that problem (7) has an strict saddle point at $(\mathbf{x}_0, \mathbf{y}_0) = ([0, 0, 0]^\top, [0, 0]^\top)$. The initial point $(\mathbf{x}_1, \mathbf{y}_1)$ is close to $(\mathbf{x}_0, \mathbf{y}_0)$ and $(\mathbf{x}_2, \mathbf{y}_2)$ is far from $(\mathbf{x}_0, \mathbf{y}_0)$. We compare the proposed algorithm MCN with GDA. The learning rate of GDA and AGD step in MCN is selected from $\{c \cdot 10^{-i} : c \in \{1, 5\}, i \in \{1, 2, 3\}\}$. For MCN method, we choose $M = 10$. We compare the running time against $P(\mathbf{x}) - P^*$, $\|\nabla P(\mathbf{x})\|_2$ and $\lambda_{\min}(\nabla^2 P(\mathbf{x}))$ for two algorithms and plot the results in Figure 1. From the curves corresponding to initial point $(\mathbf{x}_2, \mathbf{y}_2)$, we observe that both MCN and GDA converge to the minimum when the initial point is far from the strict saddle point, but MCN converges much faster than GDA. When the initial point is close to the strict saddle point, Figure 4(b) shows that the GDA algorithm gets stuck at the strict saddle point since its Hessian minimum eigenvalue are always negative. However, our MCN algorithm can reach the points which have positive Hessian minimum eigenvalues.

## 5.2 Domain Adaptation

The Domain-Adversarial Neural Network (DANN) [9] is a classic method to domain adaptation. Suppose the source domain dataset is $\mathcal{S} = \{(\mathbf{a}_i^S, b_i^S)\}_{i=1}^{N_S}$ where $\mathbf{a}_i^S$ is the feature vector of the $i$-th sample and $b_i^S$ is the corresponding label. The target domain dataset $\mathcal{T} = \{\mathbf{a}_i^\mathcal{T}\}_{i=1}^{N_\mathcal{T}}$ only contains features. Then DANN aims to solve the following nonconvex-strongly-concave minimax problem

$$\min_{[\mathbf{x}_1; \mathbf{x}_2] \in \mathbb{R}^{d_x}} \max_{\mathbf{y} \in \mathbb{R}^{d_y}} L_1(\mathbf{x}_1, \mathbf{x}_2) - \alpha \cdot L_2(\mathbf{x}_1, \mathbf{y}),$$

where $L_1(\mathbf{x}_1, \mathbf{x}_2) = \frac{1}{N_S} \sum_{i=1}^{N_S} l(\mathbf{x}_2; \Phi(\mathbf{x}_1; \mathbf{a}_i^S), b_i^S)$ is the loss of supervised learning and

$$L_2(\mathbf{x}_1, \mathbf{y}) = \frac{1}{N_S} \sum_{i=1}^{N_S} D_{\mathcal{S}}(h(\mathbf{y}; \Phi(\mathbf{x}_1; \mathbf{a}_i^S))) - \frac{1}{N_\mathcal{T}} \sum_{i=1}^{N_\mathcal{T}} D_{\mathcal{T}}(h(\mathbf{y}; \Phi(\mathbf{x}_1; \mathbf{a}_i^\mathcal{T}))) + \lambda \|\mathbf{y}\|^2$$

is the domain classification loss. Here $\Phi$ is a single-layer neural network of size $(28 \times 28) \times 200$ with parameter $\mathbf{x}_1$ and $l$ is a two-layer neural network of size $200 \times 20 \times 10$ with parameter $\mathbf{x}_2$, followed

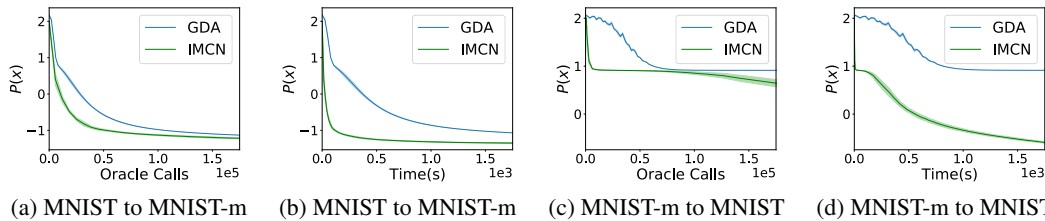

| (a) MNIST to MNIST-m | (b) MNIST to MNIST-m | (c) MNIST-m to MNIST | (d) MNIST-m to MNIST |

Figure 2: We present results on DANN model. Figure (a) and (b) show the results of domain adaptation from MNIST to MNIST-m. Figure (c) and (d) show the results of domain adaptation from MNIST-m to MNIST.

by a cross entropy loss. We choose the sigmoid function as the activation function for them. With the commonly used logistic loss for $L_2$, we let $h(\mathbf{y}; \mathbf{z}) = 1/(1 + \exp(-\mathbf{y}^\top \mathbf{z}))$, $D_S(z) = 1 - \log(z)$ and $D_T(z) = \log(1 - z)$.

Since the dimension of the minimax problem is quite large, we implement the IMCN algorithm instead of the MCN algorithm for efficiency. We compare IMCN with GDA on the domain adaptation problem between two different datasets: MNIST [19] and MNIST-m [9]. Since we do not know the close form of $P(\mathbf{x})$, we estimate the value of $P(\mathbf{x}) = \max_\mathbf{y} f(\mathbf{x}, \mathbf{y})$ by AGD procedure. More details about our experimental setup can be found in Appendix G.2.

We compare IMCN and GDA algorithms via running time and oracle calls and show the results in Figure 2. We run the experiments for five times with different random initialization and report the average results. The oracle calls of GDA only contains gradient while the oracle calls of IMCN contains both gradient and Hessian-vector product. Based on Figure 2, we observe that IMCN significantly outperforms GDA in both time and oracle comparison. Notice that IMCN requires to call much more gradient/Hessian-vector oracles on y than the gradient oracles on $\mathbf{x}$. These results verify our convergence analysis and show the advantage of proposed algorithm.

## 6 Conclusions and Future Work

In this paper, we study second-order optimization methods for nonconvex-strongly-concave minimax problems. We have proposed a novel algorithm so-called minimax cubic Newton (MCN) which could find an $(\varepsilon, \kappa^{1.5}\sqrt{\rho\varepsilon})$-SSP of the primal function with $\mathcal{O}(\kappa^{1.5}\sqrt{\rho}\varepsilon^{-1.5})$ second-order oracle calls and $\tilde{\mathcal{O}}(\kappa^2\sqrt{\rho}\varepsilon^{-1.5})$ gradient oracle calls. We also provide an efficient algorithm for high dimensional problem, which avoids accessing second-order oracle and contains $\tilde{\mathcal{O}}(\kappa^{1.5}\ell\varepsilon^{-2})$ Hessian-vector oracle calls and $\tilde{\mathcal{O}}(\kappa^2\sqrt{\rho}\varepsilon^{-1.5})$ gradient oracle calls. To best of our knowledge, this paper first achieves non-asymptotic convergence result for finding SSP of minimax problem without convex-concave assumption.

There are several interesting problems for future work: (a) The proposed algorithms and analysis require the strongly convexity assumption on $\mathbf{y}$. We would like to study how to find SSPs for general nonconvex-concave minimax problems. (b) The upper complexity bounds of proposed algorithms look not optimal. It is possible to apply the acceleration techniques to establish more efficient algorithms for our task. (c) The implementations of proposed IMCN still require accessing the Hessian-vector oracle. It is interesting to investigate how to find SSPs of our minimax problem by pure first-order algorithms. (d) This paper does not consider the specific structure of the objective function. However, many machine learning models can be formulated as minimax problems where the objective functions have finite-sum or expectation form. Designing efficient stochastic algorithms for such formulations is an interesting problem to the machine learning community.

## Acknowledgements

Luo Luo is supported by National Natural Science Foundation of China (No. 62206058) and Shanghai Sailing Program (22YF1402900). Cheng Chen is supported by Singapore Ministry of Education (AcRF) Tier 1 grant RG75/21.

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
