# OpenReview forum: "Finding Second-Order Stationary Points in Nonconvex-Strongly-Concave Minimax Optimization"
_NeurIPS.cc/2022/Conference — NeurIPS 2022 Accept_

### Official Review · Reviewer_KK4D · 2022-07-10

**Rating:** 6
**Confidence:** 4
**Soundness:** 4 excellent
**Presentation:** 3 good
**Contribution:** 3 good

**Summary:**

This work studies the nonconvex-stronglyconcave minimax optimization and proposed an algorithm called Minimax Cubic-Newton (MCN), which finds an $\mathcal{O}\left(\varepsilon, \kappa^{1.5} \sqrt{\rho \varepsilon}\right)$-second-order stationary point of $P(\mathbf{x}) \equiv \max_{\mathbf{y}\in \mathbb{R}^d} f(\mathbf{x}, \mathbf{y})$ in a query complexity of $\mathcal{O}\left(\kappa^{1.5} \sqrt{\rho} \varepsilon^{-1.5}\right)$ of second-order oracle and $\tilde{\mathcal{O}}\left(\kappa^{2} \sqrt{\rho} \varepsilon^{-1.5}\right)$ of first-order oracle, where $\kappa$ is the condition number on $\mathbf{y}$ and $\rho$ is the Lipschitz constant of $\nabla^{2} f(\mathbf{x}, \mathbf{y})$. This work also proposed a first-order approximation variant of MCN and finds an approximate second-order stationary point in a reduced query complexity of $\mathcal{O}\left(\kappa^{1.5} \ell \varepsilon^{-2}\right)$ of Hessian-vector oracle and $\tilde{\mathcal{O}}\left(\kappa^{2} \sqrt{\rho} \varepsilon^{-1.5}\right)$ first-order oracle.

The authors also discuss the concurrent work of [7] who also study the nonconvex-strongly-convex minimax problem and propose essentially the identical algorithm called Cubic-GDA, adopting a different output rule, which is helpful for the development of their stochastic Cubic-GDA. Two works have substantially different parts: this work studies the inexact variant of MCN but does not appear in [7], and [7] studies the stochastic setting which does not appear in this work.


**Questions:**

(I) Are there any lower bound arguments or results available, justifying (or negating) the optimality in complexities presented in this work and also the concurrent work [7]? Would the author mind briefly discussing this in your rebuttal at least?

(II) Is the extension to either expectation or finite-sum case straightforward? Some remarks on this would be helpful.

**Limitations:**

This is a theoretical work and does not pose limitations to my best understanding.

**Strengths And Weaknesses:**

Strength:

The first part of the analysis relies on the objective function for minimization, $P(x)=\max_y f(x,y)$, having its first-order and second-order Lipschitz constants specified well in earlier literature [20] [32]. The main complexity result, Corollary 1, provides a result that is of $\tilde{O}(\varepsilon^{-1.5})$ in both first-order and second-order query complexities, where I ignored here the $\kappa$ and $\rho$ dependencies in the $O(\cdot)$ and $\tilde{O}(\cdot)$. It suffices to combine the complexity analysis of Theorem 1 on cubic regulasization using approximate oracles [36] and also Theorem 2 on fastly achieving this approximation. To be specific, Theorem 1 follows from the analysis of the standard cubic regularised Newton’s method to achieve an $O(\varepsilon^{-1.5})$ iteration complexity via an approximate gradient and Hessian estimator. Theorem 2 applies Nesterov's AGD to find the total complexity for the ascent steps of $\tilde{O}(\varepsilon^{-1.5})$ query complexity.

This work also details the query complexity, space complexity, and flops involved in the complexity analysis. The author also introduces the concept of strict-minimax property of a nonconvex-stronglyconvex function, which is a nice generalization to the strict-saddle condition in saddle-point escaping literature [10] [16]. This is the first introduction of such a concept to my best knowledge. The mathematical derivations seem solid.

I especially enjoyed reviewing the second part, from which I learned a lot myself. Although the matrix Chebyshev polynomials approximating the inverse Hessian is not new and is ubiquitous in numerical optimization (see, e.g., the blog post \url{https://francisbach.com/chebyshev-polynomials/}), the introduction of such a method for solving the current problem using first-order oracles and Hessian-vector product oracles is considered innovative. The method for solving the inexact setting is written clearly and organically combined into the problem solution, leading to the complexity of $O(\varepsilon^{-2})$ of Hessian-vector oracle and $\tilde{O}(\varepsilon^{-1.5})$ first-order oracle.

The presentation of this work is considered clear. This work will add value to the optimization community.


Weakness:

The discussion on the optimality of the method seems unavailable. Consider the case where the function is constant in $y$, the condition number $\kappa$ on the $y$ variable is equal to 1, and the upper and lower bounds are closely matched, c.f. Carmon et al. (2020), "Lower bounds for finding stationary points I". For the general minimax problem, a lower bound argument for finding a second-order stationary point would add strength to this work, even if it is matched by an upper bound only under a nontrivial subset of the problem-dependent regimes. In addition, the discussion on the extension to a stochastic setting (either finite-sum or expectation) seems unavailable. Given the state of affairs, I encourage the authors to satisfactorily address these two points.

======

After rebuttal: I acknowledge the contribution of this work and responses during the rebuttal period, and there are some valuable discussions regarding the optimality (and suboptimality) of the provided MCN (and IMCN) methods in general. Hence I decide to raise my rating to 6.

---

> ### Author Response · Authors · 2022-08-02
> **Response to Reviewer KK4D**
>
> Thank you for your thoughtful comments and suggestions. We give answers to each question in turn.
>
> > Are there any lower bound arguments or results available, justifying (or negating) the optimality in complexities presented in this work and also the concurrent work [7]? Would the author mind briefly discussing this in your rebuttal at least?
>
> To the best of our knowledge, no existing works discuss the lower bounds for finding the second-order stationary point of nonconvex-strongly-concave minimax problem. The lower bound of our problem may be established by integrating the ideas of Han et al. [13], Zhang et al. [42] and the following paper
>
> Yair Carmon, John C. Duchi, Oliver Hinder, and Aaron Sidford. Lower bounds for finding stationary points I. Mathematical Programming 184.1 (2020): 71-120.
>
> > Is the extension to either expectation or finite-sum case straightforward? Some remarks on this would be helpful.
>
> We think the extension to either expectation or finite-sum case is possible. There are some possible ideas:
>
> 1. For expectation case, we think it's possible to use the sub-sampled gradient and Hessian to replace the exact ones in the iteration on $\bf x$, and use SGD (or its variants) to update $\bf y$ to avoid exact gradient calls.
> This idea can be straightforward applied to both MCN and inexact MCN.
>
> 2. For finite-sum case, we think it's possible to use the technique of variance reduction to construct the estimators of gradient and Hessian (this technique is also applicable to expectation case under average-smooth assumption).
> We believe such idea can obtain some interesting results of stochastic oracle complexity, but the design and analysis of variance reduced algorithms look non-trivial.

---

> > ### Comment · Reviewer_KK4D · 2022-08-07
> > **More on lower bound**
> >
> > Thank you for your answering the questions in detail. On matching the lower bounds, I do have a follow-up question: although there is no existence of lower bounds for finding a second-order stationary point of nonconvex-strongly-concave minimax problem, can you sketch a quick argument that this method indeed matches the lower bound for this problem (or even to problem-dependent constant esp. the dependency on $\kappa$)? Or it only matches the problem in a case of $\kappa=1$ (for the exact variant)?

---

> > > ### Author Response · Authors · 2022-08-07
> > > **Response to Reviewer KK4D**
> > >
> > > Thanks for raising this insightful question. We guess our algorithms may not match the lower bound. Here we provide some non-rigorous discussions on how to improve our bounds.
> > >
> > > 1. For the dependency on $\kappa$, Lin et al. [21] use the inexact proximal point method to find first-order stationary point, which has $\sqrt{\kappa}$ dependency on the number of gradient calls. We think it is possible to use the inexact high-order proximal point method [A] to reduce the dependency on $\kappa$ for finding second-order stationary point.
> > > However, the detailed design and analysis for such algorithm are non-trivial because solving the sub-problem of high-order proximal point iteration is much more difficult than Lin et al.'s [21]. We think this is an interesting open problem and would like to study it in the future.
> > >
> > > 2. The dependency on $\varepsilon$ may also be improved. Consider the objective function has the form of $f({\bf x},{\bf y})=f_1({\bf x}) - ||{\bf y}||^2$, where $f_1$ is possibly nonconvex. Then finding the approximate second-order stationary point of the minimax problem reduces to finding the approximate second-order stationary point of nonconvex function $f_1$, which can be solved by negative
> > > curvature descent with $\tilde{\mathcal{O}} (\varepsilon^{-7/4})$ number of gradient/Hessian-vector calls [B], but our IMCN algorithm requires $\tilde{\mathcal{O}} (\varepsilon^{-2})$ gradient/Hessian-vector calls. We think it's possible to improve the dependency on $\varepsilon$ by taking advantage of Yair Carmon et al.'s idea, but performing negative curvature descent on the Hessian of $P({\bf x})=\max_{\bf y} f({\bf x}, {\bf y})$ is more complicated than the case of nonconvex minimization. Overall, we think it is an interesting direction for future work.
> > >
> > > [A] Yurii Nesterov. Inexact accelerated high-order proximal-point methods. Mathematical Programming, 2021.
> > >
> > > [B] Yair Carmon, John C. Duchi, Oliver Hinder, and Aaron Sidford. Accelerated methods for non-convex optimization. SIAM Journal on Optimization, 2018.

---

> > > > ### Comment · Reviewer_KK4D · 2022-08-07
> > > > **Raising my score**
> > > >
> > > > Thank you for the answer! In the inexact case where IMCN is called, the lower bound's dependency on $\varepsilon$ is obviously not matched due to the nonaccelerated nature of the algorithmic design. Due to the authors' hard work on the rebuttal and the valuable discussions therein, I have raised my rating to 6.

---

### Official Review · Reviewer_btzP · 2022-07-11

**Rating:** 7
**Confidence:** 3
**Soundness:** 3 good
**Presentation:** 4 excellent
**Contribution:** 3 good

**Summary:**

Paper proposes and analyzes algorithms for solving for inexact second-order stationary points of nonconvex-stronglyconcave minimax problem. The first algorithm MCN adapts the known Cubic Newton method for nonconvex minimization for solving the Primal problem. This is done by generalizing the concept of strict-saddle in nonconvex minimization to nonconvex-stronglyconcave minimax problems. Since we don’t have access to gradient and Hessian of the Primal function, MCN approximates them by applying AGD on the maximization variable and the authors theoretically prove that this suffices.

However this algorithm still requires matrix inversions due to the complicated Primal Hessian. The second algorithm IMCN approximates the matrix inversion using Hessian vector products and Chebyshev polynomials. Further it uses inexact cubic solver. Authors provide theoretical analysis of the run-time and space complexities of the algorithm and provide some experimental evaluation of their methods.

**Questions:**

1. Please explain why the Algorithm 4 can’t be used in lieu of Algorithm 5 in IMCN
2. In the synthetic experiments, it is not clear why we are looking at P(x) - P^* instead of the gradient norm and Hessian minimum eigenvalue.
3. In domain adaptation experiments, can the authors also plot the final metrics for the tasks?
4. Please run the experiments over multiple random initialization and then provide mean and error bars in the plot.
5. Please summarize and cite the past work as given in [21] and all the follow up works

Minor:

5. Is "unexpected FSP" a standard usage? Should you replace all "unexpected FSP" with “strict saddle point”?
6. In Thm 1 what is s_t? Not defined yet.

**Limitations:**

Not a lot of discussion of the limitations of the work. Authors could try to discuss the challenges to the following
1. Extending to nonconvex-concave setting
2. Simplifying the algorithm to fewer loops and fewer sub-procedures

**Strengths And Weaknesses:**

*Strengths*
1. Clean idea of adapting known second-order algorithms and results in nonconvex optimization to nonconvex-stronglyconcave setting.
2. Great presentation of the main ideas of the algorithm and the proof.
3. Comprehensive presentation of the run-time and space complexities of the algorithms
4. Provides an inexact method improving the run-time and space complexities of MCN by
5. Some experimental evaluation
6. Even though I didn’t read the carefully proof, the theoretical results passes spot checks

*Weaknesses*
1. Complicated multi-loop/multi-procedure algorithm
2. Limited novelty in the algorithm, but this shouldn’t be a reason for not accepting
3. There are a few concerns with the experimental evaluation (see below)
4. Missing too many prior work on nonconvex-concave and non convex-strongly-concave optimization (see below)

*Final*

I appreciate the discussion with the authors. I keep my score the same.

---

> ### Author Response · Authors · 2022-08-02
> **Response to Reviewer btzP**
>
> Thank you for your thoughtful comments and suggestions. We give answers to each question in turn.
>
> > Please explain why the Algorithm 4 can’t be used in lieu of Algorithm 5 in IMCN.
>
> The main difference between Algorithm 4 and Algorithm 5 is that we don't need to perfrom the loop in Algorithm 4 when the norm of the gradient $g_t$ is large enough (see line 2 of Algorithm 4). Actually, if the gradient is large enough, the Cauchy-step in line 3-4 of Algorithm 4 can provide a sufficient descent for $P(\bf x)$. On the other hand, line 11-12 of Algorithm 3 aims to compute the desired approximate second-order stationary point, thus we have to solve the cubic sub-problem with a higher precision by Algorithm 5.
>
> > In the synthetic experiments, it is not clear why we are looking at $P({\bf x}) - P^*$ instead of the gradient norm and Hessian minimum eigenvalue.
>
> We plot $P({\bf x}) - P^*$ because it reflect how the algorithm converges to the optimal solution directly. We agree that plotting the gradient norm and Hessian minimum eigenvalue is also useful for understanding the advantage of our algorithm. We have added the results of these measures in the revision (Appendix G.1 of the full version in supplementary materials).
>
> > In domain adaptation experiments, can the authors also plot the final metrics for the tasks?
>
> Our experiments focus on verifying theoretical guarantees and studying the numerical performance of proposed optimization algorithms. We do not care the final metrics since they heavily depend on the  architecture and hyper-parameters of the neural networks.
>
> > Please run the experiments over multiple random initialization and then provide mean and error bars in the plot.
>
> Thank you for you suggestions. We present the results of running the experiments over five times random initialization in the revision (see Figure 2). The solid curves represent the means and the shaded regions represent the standard deviation.
>
> > Please summarize and cite the past work as given in [21] and all the follow up works.
>
> Thank you for you suggestion. We will provide a more detailed summary and cite more works related to [21]. To avoid the confusion of changing the number of reference, I prefer to do it after the stage of rebuttal.
>
> > Is "unexpected FSP" a standard usage? Should you replace all "unexpected FSP" with “strict saddle point”?
>
> Thanks for the reviewer's suggestion. In revision, we have replaced "unexpected FSP" with “strict saddle point” in Section 5 (line 288). We tend to use the informal phrase "unexpected FSP" in Section 1 (line 36), since "strict saddle point" first be  formally defined in Section 2.
>
> > In Thm 1 what is ${\bf s}_t$? Not defined yet.
>
> I guess the reviewer refer to Theorem 2 (there is no ${\bf s}_t$ in Theorem 1). The ${\bf s}_t$ in Theorem 2 should be ${\bf s}^*_t$. We have fixed it in the revision.
>
> > discuss the challenges of "Extending to nonconvex-concave setting"
>
> For nonconvex-concave setting, the primal function $P({\bf x})$ could be non-smooth. One option is studying its Moreau envelope $P_\lambda({\bf x})=\min_{\bf x} P({\bf x})+\frac{1}{2\lambda}||{\bf x}||^2$ [Appendix A, 21] to characterize the local optimality.
>
> > discuss the challenges of "Simplifying the algorithm to fewer loops and fewer sub-procedures"
>
> It is possible to run only one step of AGD iteration at each round (set $K_t=1$ for all $t$). We are interested to study whether such simplification can still keep existing upper bound complexity. Simplifying the inexact cubic sub-problem solver looks non-trivial, but we believe it is a good future direction.

---

> > ### Comment · Reviewer_btzP · 2022-08-08
> > **Reply**
> >
> > Thank your for the response. I also appreciate the new plots! I just one follow up request.
> >
> > > We do not care the final metrics since they heavily depend on the architecture and hyper-parameters of the neural networks.
> >
> > This is very weird statement to make in a ML conference review. Readers might care about the generalization performance of model. This is especially true since it is known that the there is a known relation between the spectrum of the Hessian and generalization performance and this algorithm specifically selects iterates based on Hessian spectrum. Therefore, I hope the authors can provide such a plot.
> >
> > Minor:
> > > One option is studying its Moreau envelope
> >
> > Formula of Moreau envelope is wrong.

---

> > > ### Author Response · Authors · 2022-08-09
> > > **Response to the follow up questions**
> > >
> > > Thank you for your feedback.
> > >
> > > > there is a known relation between the spectrum of the Hessian and generalization performance
> > >
> > > Thank you for your pointing out the relation between the generalization performance and the Hessian spectrum. Due to the time limitation, we do not have enough time to run all the domain adaptation experiments and test the final metrics in the author-reviewer discussion period. We will try to provide the plots of the final metrics in later revision.
> > >
> > > > Formula of Moreau envelope is wrong.
> > >
> > > Thanks for the minor comment. The Moreau envelope should be $P_\lambda({\bf x})=\min_{\bf w} P({\bf w})+\frac{1}{2\lambda}||{\bf w - \bf x}||^2$.

---

### Official Review · Reviewer_JH7L · 2022-07-13

**Rating:** 5
**Confidence:** 3
**Soundness:** 4 excellent
**Presentation:** 4 excellent
**Contribution:** 2 fair

**Summary:**

This is paper studies the problem of finding second-order stationary points (SSP) of Nonconvex-Strongly-Concave (NC-SC) minimax optimizaiton problems. The propose an algorithm called Minimax Cubic Newton (MCN) and show it provably converges to an approximate SSP and provide complexity bounds of calling first and second order oracles. The bounds are in terms of the condition number $\kappa$, Lipschitz constant of Hessian $\rho$ and level of stationarity $\epsilon$. Next, to avoid call expensive second-order oracles, they further propose inexact MCN which only needs the less expensive Hessian-vector oracle. The also provide convergence rate for inexact MCN and finally conducted experiments to show their algorithm is better than GDA.

**Questions:**

Question:
1. How good is the proposed algorithm and the corresponding rate? Are there any lower complexity bounds for the problem of interest?

Minor comments:
1. Some theorems look a bit complicated, maybe the authors could consider using more $\tilde{O}$ to hide log factors.
2. It might be better to use a larger font for the figures.

**Limitations:**

I do not think this paper has any potential negative societal impact. Regarding limitations, please see the weakness part.

**Strengths And Weaknesses:**

Strengths:
1. The theoretical results are sound and look rigorous. The presentation is clear and easy to read. They also  provide clear intuition of how the results are shown.
2.  They are the first one to study convergence to SSP for minimax optimization problems without convex-concave assumptions, which is novel.

Weaknesses:
1. The paper looks not novel enough. The exact MCN algorithm looks pretty natural, it looks like a combination of Accelerated Gradient method for the inner maximization problem and the Cubic Regularized Newton method for the outer minimization problem. The analysis seems also a combination. However, the inexact MCN and its analysis looks novel and interesting to me.

---

> ### Author Response · Authors · 2022-08-02
> **Response to Reviewer JH7L**
>
> Thank you for your thoughtful comments and suggestions. We give answers to each question in turn.
>
> > The paper looks not novel enough. The exact MCN algorithm looks pretty natural, it looks like a combination of Accelerated Gradient method for the inner maximization problem and the Cubic Regularized Newton method for the outer minimization problem. The analysis seems also a combination.  However, the inexact MCN and its analysis looks novel and interesting to me.
>
> We agree that the main theoretical challenge in our paper is the analysis of inexact MCN. However, the MCN method can be applied to low-dimensional problems and it could help us understand the implementation and the analysis of inexact MCN.
>
> >  How good is the proposed algorithm and the corresponding rate? Are there any lower complexity bounds for the problem of interest?
>
> The existing lower bounds for nonconvex-strongly-concave minimax problem only consider finding the first-order stationary point [13, 42]. To the best of our knowledge, no existing works discuss the lower bounds of finding the second-order stationary points of such minimax problem yet. The lower bound of our problem may be established by integrating the ideas of Han et al. [13], Zhang et al. [42] and Carmon et al.'s paper. It is interesting to study whether the complexity bounds obtained in this paper can be improved further.
>
> > 1. Some theorems look a bit complicated, maybe the authors could consider using more $\tilde{O}$ to hide log factors.
> 2. It might be better to use a larger font for the figures.
>
> Thanks the reviewer's suggestion for presentation. Considering it may potentially affect the layout of our paper, we prefer to do such adjustment after the stage of rebuttal.

---

> > ### Comment · Reviewer_JH7L · 2022-08-08
> > **Thank you for answering my questions!**
> >
> > Thank you for your answers to my questions! However, after reading all reviewers' comments and the authors' replies, I still want to keep my score at 5.

---

### Official Review · Reviewer_byDE · 2022-07-19

**Rating:** 6
**Confidence:** 3
**Soundness:** 3 good
**Presentation:** 3 good
**Contribution:** 3 good

**Summary:**

This paper mainly studied minimax optimization problem. It proposes an algorithm MCN to find SPP and also provides an algorithm for high dimensional problem. Numerical experiments are also conducted to validate the theory.

**Questions:**

See "Strengths and Weaknesses"

**Limitations:**

See "Strengths and Weaknesses"

**Strengths And Weaknesses:**

1. The paper emphasizes that this is the first work without the convex-concave assumptions. Could you give more comparison with prior literature and explanation on how this is accomplished technically?

2. The theory of this paper is developed under a group of assumptions. Could the author give more justifications on the practicality of the assumptions?

3. The paper is generally clear and well-written.

---

> ### Author Response · Authors · 2022-08-02
> **Response to Reviewer byDE**
>
> Thank you for your thoughtful comments and suggestions. We give answers to each question in turn.
>
> > The paper emphasizes that this is the first work without the convex-concave assumptions. Could you give more comparison with prior literature and explanation on how this is accomplished technically?
>
> Our paper is the first work that considers non-asymptotic convergence behavior of finding second-order stationary point for minimax problems without convex-concave assumptions.
> The existing non-asymptotic convergence analyses for minimax problem without convex-concave optimization mainly focus on finding first-order stationary point (FSP), see [13, 20, 21, 42]. These methods are gradient-based and do not consider the second-order information.
> However, a FSP of a nonconvex-concave minimax problem may not have good local optimality. Thus we study how to find a second-order stationary point (SSP) for nonconvex-concave minimax problems.
>
> The main idea of our work is applying the (approximate) cubic regularized Newton method on $\bf x$ and AGD on $\bf y$. We leverage the second-order information to guarantee that the Hessian of the primal function $P({\bf x})=\max_{{\bf y}} f({\bf x},{\bf y})$ is nearly positive semi-definite.
> For high-dimensional problems, we proposed an inexact method (IMCN, Algorithm 3) which adopts Chebyshev approximation to avoid constructing Hessian matrices and exact cubic sub-problem solver. The IMCN algorithm is practical (Appendix E) and theoretical guaranteed (Section 4.2).
>
> > The theory of this paper is developed under a group of assumptions. Could the author give more justifications on the practicality of the assumptions?
>
> All assumptions in our paper are mild. Assumption 1, 3 and 4 are commonly used in first-order algorithms for nonconvex-concave minimax optimization [20, 21]. Assumption 2 is commonly used in second-order optimization [4, 5, 6, 17]. In addition, many machine learning applications including reinforcement learning [29], domain adaptation [9] and adversarial training [33], satisfy these assumptions.

---

> > ### Comment · Reviewer_byDE · 2022-08-09
> > **Thanks for the response.**
> >
> > Thank you for the detailed response. I'll raise my rating to 6.

---

### Meta-Review · Area_Chair_JpVb · 2022-08-20

**Recommendation:** Accept
**Confidence:** Certain

**Metareview:**

This paper studies the minimax optimization problem with smooth objective function, where the objective function $f(x,y)$ is assumed to be  strongly concave in $y$ but in general nonconvex in $x$. In comparison to prior non-asymptotic results that mostly focused on finding first-order stationary points, this paper takes an important step further by showing how to find a second-order stationary point with non-asymptotic convergence guarantees. Although the algorithm design herein is somewhat straightforward (i.e., it is accomplished via a simple combination of accelerated gradient methods and cubic regularized newton methods), the analysis for inexact MCN contains sufficient novelty.  As a result, I recommend acceptance of this paper.

**Award:**

No

---

### Decision · Program_Chairs · 2022-09-14

Accept